# Genetic Variability of Mineral Content in Different Grain Structures of Bean Cultivars from Mesoamerican and Andean Gene Pools

**DOI:** 10.3390/plants10061246

**Published:** 2021-06-19

**Authors:** Douglas Mariani Zeffa, Alison Fernando Nogueira, Juliana Sawada Buratto, Raquel Barboza Reis de Oliveira, José dos Santos Neto, Vânia Moda-Cirino

**Affiliations:** 1Agronomy Department, State University of Maringá, Maringá 87020-900, Brazil; 2Agronomy Department, State University of Londrina, Londrina 86057-970, Brazil; alllisonfernando@gmail.com (A.F.N.); raquelbarbozabio@yahoo.com.br (R.B.R.d.O.); 3Plant Breeding and Propagation Area, Paraná Rural Development Institute-IAPAR-EMATER, Londrina 86047-902, Brazil; jsburatto@idr.pr.gov.br (J.S.B.); js.neto@idr.pr.gov.br (J.d.S.N.); vmvamoci@gmail.com (V.M.-C.)

**Keywords:** *Phaseolus vulgaris* L., biofortification, macronutrient, micronutrient, protein

## Abstract

Beans (*Phaseolus vulgaris* L.) are an important source of proteins, carbohydrates, and micronutrients in the diets of millions of people in Latin America and Africa. Studies related to genetic variability in the accumulation and distribution of nutrients are valuable for biofortification programs, as there is evidence that the seed coat and embryo differ in the bioavailability of essential nutrients. In this study, we sought to evaluate the genetic variability of total mineral content in the grain and its constituent parts (seed coat, cotyledon, and embryonic axis) of bean genotypes from Mesoamerican and Andean centers of origin. Grain samples of 10 bean cultivars were analyzed for the content of proteins and minerals (Mg, Ca, K, P, Mn, S, Cu, B, Fe, and Zn) in the whole grains and seed coat, cotyledons, and embryonic axis tissues. Genetic variability was observed among the cultivars for protein content and all evaluated minerals. Moreover, differential accumulation of minerals was observed in the seed coat, cotyledons, and embryonic axis. Except for Ca, which accumulated predominantly in the seed coat, higher percentages of minerals were detected in the cotyledons. Furthermore, 100-grain mass values showed negative correlations with the contents of Ca, Mg, P, Zn, Fe, and Mn in whole grains or in the different grain tissues. In general, the Mesoamerican cultivars showed a higher concentration of minerals in the grains, whereas Andean cultivars showed higher concentrations of protein.

## 1. Introduction

The common bean (*Phaseolus vulgaris* L.) is considered an important source of protein, complex carbohydrates, polyunsaturated fatty acids (linoleic and linolenic), soluble and insoluble fiber, vitamins (particularly of the B complex), and the minerals calcium, potassium, magnesium, iron, copper, and zinc [1,2,3,4]. This legume is a major food source in terms of caloric intake and nutrients worldwide, particularly in developing countries in East Africa and Latin America [5,6].

Although genetic variability in the mineral contents of different bean types worldwide has previously been reported, these analyses have generally been performed considering the whole grain and have rarely focused on contents in the different constituent grain tissues [7,8,9]. Different tissues differ with respect to the bioavailability of mineral nutrients essential to the human diet. For example, the seed coat must be considered specifically as it accumulates much of the anti-nutrients such as tannins that affect mineral bioavailability [10]. Meanwhile, the cotyledons accumulate phosphorus in the form of phytates, considered an important anti-nutritional factor [11]. In this sense, information relating to genetic variability in the accumulation and distribution of minerals in different bean grain tissues is of particular importance with respect to enhancing the increment of these minerals in breeding programs [10,11].

The grain of beans has three main components, namely, the seed coat, cotyledons, and the embryonic axis [12]. In the common bean, the cotyledons comprise approximately 90% of the total dry biomass of the grain, the seed coat 9%, and the embryonic axis 1% [13]. Minerals typically accumulate to differing levels in these three grains tissues [10,14]. Ribeiro et al. [13], who evaluated 16 common bean genotypes from the Mesoamerican and Andean gene pools, reported that more than 94.5% of the calcium present in the grains is concentrated in the seed coat, whereas more than 76.0% of the potassium is distributed in the embryo.

Bean genotypes of Mesoamerican origin are characterized by small or medium-sized seeds (25–40 g per 100 seeds), whereas those of Andean origin are somewhat larger (>40 g per 100 seeds) [15]. In general, Mesoamerican genotypes typically have higher grain mineral contents than Andean genotypes [7]. In addition, studies have reported negative correlations between 100-grain mass values and mineral contents [13,16]. Although some studies have evaluated the content and distribution of minerals in different grain tissues, there are few studies comparing genotypes of Andean and Mesoamerican origin. In this sense, our objective was to evaluate the variability of mineral contents in the seed coat, cotyledon, embryonic axis, and whole grain of beans derived from different centers of origin.

## 2. Results

### 2.1. ANOVA: Mineral and Protein Contents in Grains

The results obtained for variance analyses are presented in Table 1. Except for the content of S, cultivars showed significant effects (*p* < 0.01) for all the evaluated characteristics. There was a significant effect (*p* < 0.01) for the contents of all minerals evaluated in different grain tissues. Moreover, with the exceptions of S and P, significant effects (*p* < 0.05) were observed for the cultivar × tissue interactions of all minerals. We obtained Prot values of between 14.93 (cv. IAPAR 31) and 21.36% (cv. Jalo Precoce), whereas the average contents of macronutrients ranged from 0.226 (P in the seed coat) to 4.262 g kg^−1^ (P in the embryonic axis), and that of micronutrients ranged from 0.749 (S in the seed coat) to 112.811 mg kg^−1^ (Fe in the embryonic axis).

### 2.2. Correlation between Characters

The correlation networks obtained for the characteristics evaluated in whole grains, seed coat, cotyledons, and embryonic axis are presented in Figure 1. M100 showed significant negative correlations (*p* ≤ 0.05) with the contents of Zn (*r* ≤ −0.82) (whole grains, seed coat, cotyledons, and embryonic axis), P (*r* ≤ −0.53) (whole grains, seed coat, and cotyledons), Mn (*r* ≤ −0.51) (whole grains, cotyledons, and embryonic axis), Mg (*r* ≤ −0.80) (whole grains, seed coat, and embryonic axis), Ca (*r* ≤ −0.81) (whole grains and seed coat), and S (*r* ≤ −0.78) (seed coat). We also detected negative correlations between Prot and the minerals Mg (*r* ≤ −0.61) (whole grains, cotyledons, and embryonic axis), Ca (*r* ≤ −0.62) (whole grains and seed coat), Mn (*r* ≤ −0.52) (whole grains and cotyledons), Zn (*r* ≤ −0.56) (seed coat and embryonic axis), and K (*r* = −0.63) (seed coat).

Significant positive correlations (*p* ≤ 0.05) were observed between Zn and the minerals P (*r* ≥ 0.59) (whole grains, seed coat and cotyledons), S (*r* ≥ 0.63) (whole grains, seed coat and cotyledons), Fe (*r* ≥ 0.58) (whole grains and seed coat), Ca (*r* = 0.92) (seed coat), and Mg (*r* = 0.88) (embryonic axis). In addition, Ca and Mg (*r* ≥ 0.54) (whole grains, cotyledons, and embryonic axis) and Fe and S (*r* ≥ 0.51) (whole grains, seed coat, and cotyledons) also showed positive correlations. P and S showed positive correlations with whole grains (*r* = 0.96), seed coat *(r* = 0.51), and cotyledons (*r* = 0.97), although they were found to be negatively correlated with the embryonic axis (*r* =−0.59).

### 2.3. Mineral Content of Whole Grains

Representations of the results obtained for Ward’s clustering and principal component analysis (PCA) of the M100 and Prot traits and mineral content in whole grains are presented in Figure 2. The dendrogram separated the cultivars into two groups (Figure 2a). Group 1 was constituted by the five cultivars of Mesoamerican origin (IPR Uirapuru, BRS Gralha, IPR Juriti, IPR Siriri, and IAPAR 31) and group 2 by the cultivars of Andean origin (Jalo Precoe, RedHawk, Hooter, IPR Garça, and BRS Radiante), respectively. In general, the cultivars of Andean origin are characterized by the highest average values of Prot and M100 (Figure 2a and Appendix A), whereas those of Mesoamerican origin are characterized by the highest average grain mineral contents. The first two principal components of the PCA plot (PC1 and PC2, respectively) were found to explain 70.91% of the total variation (Figure 2b), and the results generally correspond to the patterns revealed in the dendrogram, indicating a clear distinction between the cultivars of Andean and Mesoamerican origin.

### 2.4. Mineral Content of the Seed Coat

The results obtained for Ward’s clustering and PCA analyses of the mineral contents in the grain seed coat are presented in Figure 3. Consistent with the findings for whole grains, the dendrogram separated cultivars into two groups (Figure 3a), with groups 1 and 2 being respectively constituted by the five cultivars of Andean (Jalo Precoe, Red Hawk, Hooter, IPR Garça, and BRS Radiante) and Mesoamerican origin (IPR Uirapuru, BRS Gralha, IPR Juriti, IPR Siriri, and IAPAR 31). In general, the cultivars of Andean origin are characterized by the highest average values of B and Cu (Figure 3a and Appendix A), whereas Mesoamerican cultivars were found to have the highest contents of P, Ca, Fe, Zn, and S (Figure 3a and Appendix A). The first two principal components of the PCA plot explained 74.35% of the total variation (Figure 2b), with the clustering of cultivars showing a pattern similar to that observed in the dendrogram, indicating a clear distinction between the cultivars of Andean and Mesoamerican origin.

### 2.5. Mineral Content of Cotyledons

The results of Ward’s clustering and PCA of mineral contents in the cotyledons are presented in Figure 4. In contrast to our analyses of the mineral contents in the grain seed coat, the dendrogram separated the cultivars into three different groups, with no clear distinction between the two centers of origin (Figure 4a). Group 1 (IPR Garça and BRS Radiante) was characterized by the lowest average values for the minerals P, S, and Zn (Figure 4a and Appendix A). Groups 2 and 3 were constituted by five (IAPAR 31, IPR Siriri, IPR Juriti, and IPR Uirapuru) and three (Jalo Precoce, RedHawk, and IPR Gralha) cultivars, respectively, the former of which was characterized by moderate to high values of mineral contents in the seed coat, whereas group 3 cultivars were those with low average contents of Mg, Ca, and Mn. The first two principal components of the PCA plot explained 76.09% of the total variation (Figure 4b). Notably, however, whereas the three groups revealed in the dendrogram each comprised cultivars from the two different centers of origin, a clearer distinction between Andean and Mesoamerican cultivars was observed in the PCA plot.

### 2.6. Mineral Content in the Embryonic Axis

The results of Ward’s clustering and PCA analyses of mineral contents in the embryonic grain axis are presented in Figure 5. Three groups were formed in the clustering dendrogram (Figure 5a). Groups 1 (Jalo Precoe, Red Hawk, Hooter, and BRS Radiante) and 2 (IPR Garça) were constituted by the cultivars of Andean origin, whereas group 3 comprised the five Mesoamerican cultivars (IPR Uirapuru, BRS Gralha, IPR Juriti, IPR Siriri, and IAPAR 31). Group 1 cultivars were found to be characterized by the lowest averages values for the minerals Ca, Mg, S, Fe, Mn, and Zn (Figure 5a and Appendix A), whereas the IPR Garça cultivar in group 2 has the highest average values of S and Ca, although low contents of P, B, and Cu. In general, the cultivars in group 3 were found to have the highest average values for Mg, Zn, Mn, and Fe. The first two principal components of the PCA plot explained 74.38% of the total variation (Figure 5b), and the PCA clustering is broadly consistent with that observed in the dendrogram, indicating a clear distinction between the cultivars of Andean and Mesoamerican origin.

### 2.7. Mineral Partitioning in Grains

Appendix A presents the average proportions of grain tissue biomass for the 10 cultivars evaluated, which indicates that there is little difference among these cultivars with respect to proportional biomass distribution in grain tissues. The overwhelming majority (90.4%) of grain biomass is accumulated in the cotyledons, whereas the seed coat and embryonic axis account for 8.3% and 1.3%, respectively. The average distributions of minerals in the different tissues of the grains are presented in Figure 6. Except for Ca, the highest proportion of which (72.5%) accumulates in the seed coat, and the highest amounts of minerals are distributed in the grain cotyledons. The average proportions of minerals accumulated in the seed coat varied from 2.3% (S) to 72.5% (Ca), whereas in the cotyledons, they ranged between 26.7% (Ca) to 97.1% (S), and in the embryonic axis, between 0.6% (S) to 6.5% (P). The first two principal components of the PCA plot explained 90% of the total variation (Appendix A). In general, the concentrations of Mg and Ca in the seed coat were found to be higher than those in the cotyledonary and embryonic tissues, whereas concentrations of K and S were higher in the cotyledons; the embryonic axis had the highest concentrations of P, Fe, B, Z, and Cu; and the highest concentrations of Mn accumulate in the cotyledons and embryonic axis.

## 3. Discussion

### 3.1. Genetic Variability of Mineral Contents

Gaining an understanding of the distribution of mineral contents in grains is of particular importance from the perspective of assessing the biofortification of bean crops [9,13]. In the present study, we found evidence of genetic variability with respect to the macro- and micronutrient contents in bean cultivars derived from two different centers of origin. Genetic variability is a fundamental factor underpinning breeding programs that seek to enhance mineral contents and has been reported in several bean studies worldwide [4,10,17,18]. In addition to the occurrence of genetic variability among cultivars, we also observed differences in mineral contents in the different tissues of bean grains, which is consistent with the findings of previous studies that have reported differentiation in the mineral concentrations of beans [9] and other crops [14].

### 3.2. The Gene Pool and Correlation between Characters

We established that there exists a clear distinction between the gene pools of Andean and Mesoamerican cultivars. In general, Mesoamerican cultivars were found to be characterized by higher concentrations of minerals in the grains, whereas Andean cultivars tended to have higher protein contents. In contrast, Blair [10] reported similar patterns of mineral accumulation in these two gene groups. However, on the basis of their evaluation of mineral content in a collection of 1500 bean accessions, Beebe et al. [7] found that Mesoamerican genotypes showed a tendency to accumulate higher amounts of Ca, P, S, and Zn compared with Andean genotypes, which were observed to have higher grain concentrations of Fe.

We also found that values obtained for the mass of 100 grains were negatively correlated with the contents of minerals Ca, Mg, P, Mn, Fe, and Zn in whole grains or different grain tissues, thereby indicating that the larger the grain size, the lower is the concentration of these minerals. Similar results have been obtained by Hacisalihoglu et al. [16], and Blair et al. [18] have reported differing correlations among minerals in different grain tissues, which corroborates the results obtained in the present study.

### 3.3. Macronutrients: Ca, P, Mg, and K

Among the minerals evaluated in the present study, we found Ca to be the only mineral observed in greater quantity in the seed coat than in either the cotyledons or embryonic axis. Although in the present study, we obtained an average value of 72.5% for the proportion of Ca in the seed coat of bean grains, Ribeiro et al. [13] reported that, depending on genotype, contents can be as high as 97%. Moreover, Jost et al. [19] reported additive and maternal inheritance effects on Ca concentration in bean grains, which have direct implications with respect to breeding programs, given that, unlike the cotyledons and embryonic axis, which are products of fertilization, the grain seed coat is of maternal origin [20]. Among the genotypes evaluated, we found that the IAPAR 31 cultivar stood out, being classified in those groups with the highest contents of Mg, Ca, K, and P in all three grain tissues evaluated.

Although we found that a majority of grain P is distributed in the cotyledons, the embryonic axis is characterized by the highest concentration of this element, followed by the cotyledons and seed coat. These observations are consistent with those reported by Ariza-Nietoetal et al. [21], who evaluated the mineral contents of these three grain tissues in eight bean genotypes of Andean and Mesoamerican origin. Within grains, P is stored predominantly in the form of phytic acid, which is considered one of the main anti-nutritional factors in plants, owing to its tendency to bind to other minerals and nutrients, thereby altering their digestibility and absorption by the human body [22]. Phytic acid accounts for between 65% and 85% of total P content in grains and is differentially distributed among cotyledons (95–98%), embryonic axis (1–3%), and seed coat (0.5–4%) [17,23], with that stored in the cotyledons and embryonic axis serving as a source of P for germination and development during the early stages of plant growth [24].

### 3.4. Micronutrients: Mn, S, Cu, B, Fe, and Zn

The micronutrients Cu and Mn are of considerable importance with respect to health, given that a deficiency in these minerals can contribute to retarded growth and development in children [18,25]. In addition, these two micronutrients play an important role in reducing oxidative stress [9]. In plants, B present in the cotyledons and embryonic axis is required for early development, whereas in humans, this element is essential for bone growth and multiple functions of the central nervous system [26]. As a key component of proteins, S is a fundamental element for the biosynthesis of essential amino acids such as methionine and cystine [27].

Fe and Zn are considered micronutrients of primary interest with respect to bean breeding programs that seek to enhance biofortification [17,28,29]. Nutritional deficiencies associated with low intakes of Fe and Zn are common in developing countries, being linked to a number of health problems and retarded growth, particularly in women and children [30,31]. The mean contents of Fe (42.331 mg kg^−1^) and Zn (22.960 mg kg^−1^) measured in the present study are similar to the values obtained by Chávez-Mendoza et al. [9]. Comparatively, Akond et al. [8], who quantified the mineral contents in 29 bean genotypes of Andean and Mesoamerican origin, reported values between 8.9 and 112.9 mg kg^−1^ and 30.9 and 64.6 mg kg^−1^ for Fe and Zn, respectively.

It has been established that the concentrations of Fe and Zn in grains are determined by multigenetic inheritance but are also strongly influenced by environmental factors [32,33]. In addition, the accumulation patterns of these minerals in different grain tissues have been demonstrated to be under the influence of genetic control [10]. However, given that pleiotropic quantitative trait loci responsible for the accumulation of Fe and Zn in grains have been identified, it would appear that there is a common physiological process underlying the transport and accumulation of these micronutrients in grains [32]. In the present study, we obtained proportional values 15.9% and 7.8% for the contents of Fe and Zn, respectively, in the seed coat, which are consistent with those reported by Ribeiro et al. [13]. According to Ariza-Nieto et al. [21], the bioavailability of Fe that accumulates in the cotyledons is higher than that of the seed coat content of this element; however, this differential tissue-related bioavailability was found to be genotype-dependent.

## 4. Material and Methods

### 4.1. Plant Material and Experimental Design

For the purposes of the present study, we analyzed the mineral contents and their distribution in grain samples of 10 bean cultivars, among which five are of Mesoamerican origin (IPR Siriri, Juriti, Uirapuru, Gralha, and IAPAR 31) and five are of Andean origin (Garça, Hooter, Radiante, Jalo Precoce, and Red Hawk), developed by different research institutions (Table 2). The seeds of these cultivars were obtained from the Active Germplasm Bank (BAG) of the Paraná Rural Development Institute (IDR-Paraná), Londrina, Paraná, Brazil.

The analyzed grains were collected during an experiment conducted at the IDR-Paraná Research Station in Londrina, Paraná, Brazil (23°17′34′′ S, 51°10′24′′ W, and altitude 550 m). The climate of the region is classified as Cfa (humid subtropical climate) according to the Köppen-Geiger climate classification [34]. The experimental design used was a randomized complete block design with four repetitions. The experimental units consisted of two central rows of 4 m in length, with a spacing of 0.5 m between rows and 12 plants per linear meter. At seeding, soil fertilization was performed using 300 kg ha^−1^ of compound fertilizer (N, P_2_O_5_, K_2_O; 4:30:10) and urea (40 kg N ha^−1^) applied as a covering fertilizer at the V4 stage of development [35]. Pest, disease, and weed control were performed according to the technical recommendations of the crop.

At the stage of physiological maturity stage [35], beans were manually harvested from each experimental unit. Each sample comprised 1000 grains free of mechanical or insect damage, from which we determined the mass of 100 grains (M100, in grams). Prior to further analysis, the samples were stored in a cold chamber at a temperature of 5.5 ± 0.2 °C and 33% humidity.

### 4.2. Analysis of Mineral Composition and Protein Content

Prior to laboratory analyses, the grain samples were washed to eliminate impurities in the following order: tap water, 0.01 M HCl solution, and distilled water. For fractionation of grain components, following decontamination, the grain samples were placed in containers containing 1 L of distilled water for a 60 min soaking. Subsequently, the grains were separated into the seed coat, cotyledons, and embryonic axis, which, along with whole grains, were dried separately in an oven for 48 h at 60 °C. The dried samples thus obtained were ground in a Willey MA340 mill (Marconi, Piracicaba, Brazil), and the respective flours were packed in glass containers.

For the determination K, P, Ca, Mg, S, Cu, Zn, B, Mn, and Fe contents, the samples were digested in nitro-perchloric solution (HNO_3_:HClO_4_) in accordance with the methodology described by Miyazawa et al. [36], with values being obtained using an atomic emission spectrophotometer (ICP-AES) (Thermo Jarrell Ash ICAP61E; Thermo Scientific, Waltham, MA, USA). Values of the macronutrients K, P, Ca, Mg, and S are expressed as g nutrient per kg dry matter (DM), whereas those of the micronutrients Cu, Zn, B, Mn, and Fe are expressed in terms of mg nutrient per g DM.

Protein content (Prot, in %) was quantified in whole-grain meal samples using the Kjeldahl method, with readings being obtained using an Agilent 8453 Ultra-Violet-Vis spectrophotometer (Agilent Technologies, Santa Clara, CA, USA). A factor of 6.25 was applied to convert total nitrogen into crude protein.

### 4.3. Statistical Analyses

Analyses of variance (ANOVA) of the characteristics M100, Prot, and mineral contents in whole grains were performed using the following mathematical model:(1)$$Y_{ij\;}=\text{µ}+c_{i}+b_{j}+\varepsilon_{ij}$$
where *Y_ij_* is the observed value of the experimental unit containing cultivar *i* in block *j*; *µ* is the general average; *c_i_* is the effect of cultivar *i*; *b_j_* is the effect of block *j*; and *ε_ij_* is the experimental error. ANOVA in split-plot schemes for mineral contents in the different grain components was performed using the following model:(2)$$Y_{ijk\;}=\text{µ}+c_{i}+t_{j}+b_{k}+\varepsilon_{ik}+ct_{ij}+\varepsilon_{ijk}$$
where *Y_ijk_* is the observed value of the experimental unit containing cultivar *i*, seed tissue *j*, and block *k*; *µ* is the overall mean; *c_i_* is the effect of cultivar *i*; *t_j_* is the effect of seed tissue *j*; *b_k_* is the effect of block *k*; *ε_ik_* is the error associated with the plots (error a); *ct_ij_* is the effect of the interaction between cultivar *i* and tissue *j*; and *ε_ijk_* is the error associated with the subplots (error *b*). The means of the treatments were grouped using the Scott–Knott test [37] at 5% probability level.

The relationship between characteristics was verified by simple Pearson linear correlation analysis using the correlation network approach. The phenotypic divergence between cultivars of Andean and Mesoamerican origin was verified by conducting principal component analysis (PCA) and Ward’s hierarchical clustering [38], based on the standardized mean Euclidean distance. The ideal number of groups formed in the dendrograms was established using the Mojena’s method [39]. Statistical analyses were performed with R software (https://www.r-project.org/) using the “Exp.Des” [40], “ggplot2” [41], “factoextra” [42], “pheatmap” [43], and “qgraph” [44] packages.

## 5. Conclusions

In the present study, we established that there is genetic variability with respect to the protein and mineral contents of 10 bean cultivars of Andean and Mesoamerican origin. Furthermore, we identified the differential accumulation of minerals in the seed coat, cotyledons, and embryonic axis of these beans. With the exception of Ca, which is predominantly concentrated in the seed coat, minerals were found to accumulate to greater extents in the grain cotyledons. We also detected negative correlations between 100-grain mass values and contents of the minerals Ca, Mg, P, Zn, Fe, and Mn in whole grains or in the different grain tissues, indicating that the cultivars of Andean origin characterized by a larger grain size have lower contents of these minerals. In contrast, however, the Andean cultivars were found to have higher grain protein contents. Among the 10 cultivars assessed, IAPAR 31 is of particular note, given that it was classified in those groups with the highest mean contents of Mg, Ca, K, and P in the three examined grain tissues.

## Figures and Tables

**Figure 1 plants-10-01246-f001:**
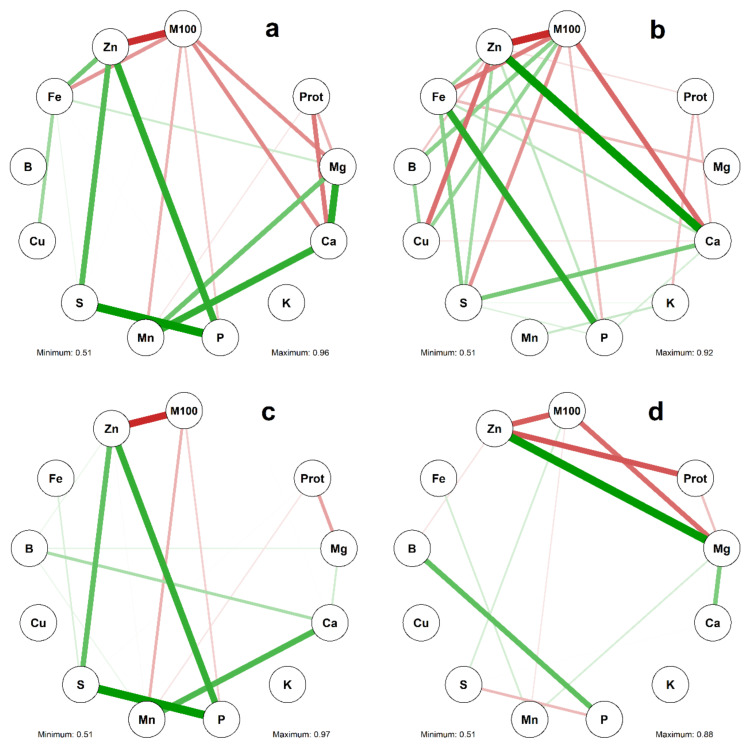
Networks of correlations between the characteristics of 100-grain mass (M100), protein content (Prot), and contents of 10 mineral in the whole grains (**a**), seed coat (**b**), cotyledons (**c**), and embryonic axis (**d**) of 10 bean (*Phaseolus vulgaris* L.) cultivars derived from different centers of origin. The green and red lines represent estimates of the positive and negative simple Pearson linear correlations (*p* ≤ 0.05), respectively. The thickness of the lines is proportional to the magnitude of the correlation.

**Figure 2 plants-10-01246-f002:**
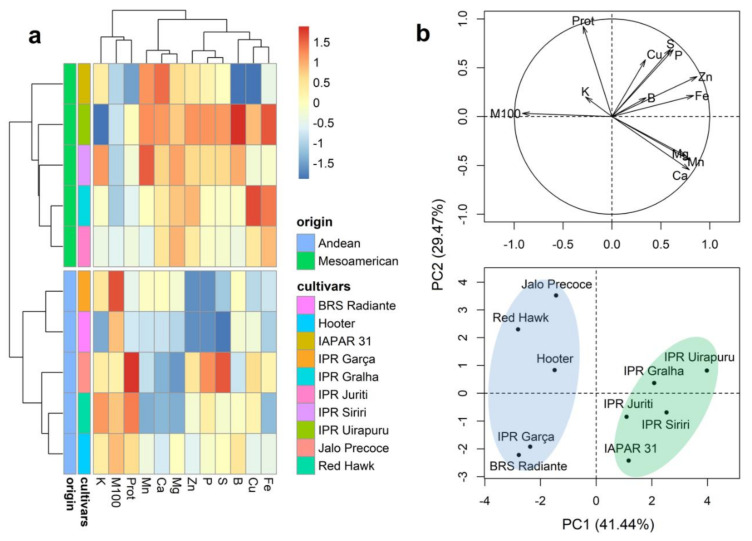
Dendrogram obtained based on Ward’s method (**a**) and principal component analysis (**b**) of 10 bean (*Phaseolus vulgaris* L.) cultivars evaluated for the characteristics 100-grain mass (M100), protein content (Prot), and the contents of 10 minerals in whole grains.

**Figure 3 plants-10-01246-f003:**
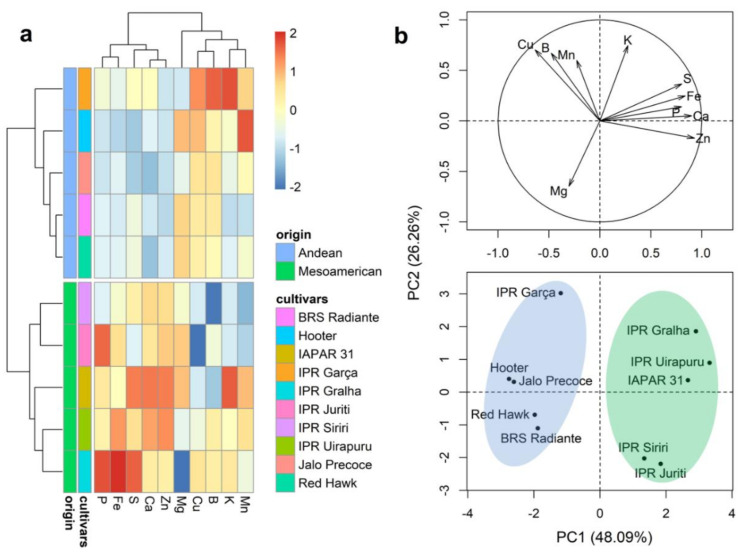
Dendrogram obtained based on Ward’s method (**a**) and principal component analysis (**b**) of 10 bean (*Phaseolus vulgaris* L.) cultivars evaluated for the contents of 10 minerals in the grain seed coat.

**Figure 4 plants-10-01246-f004:**
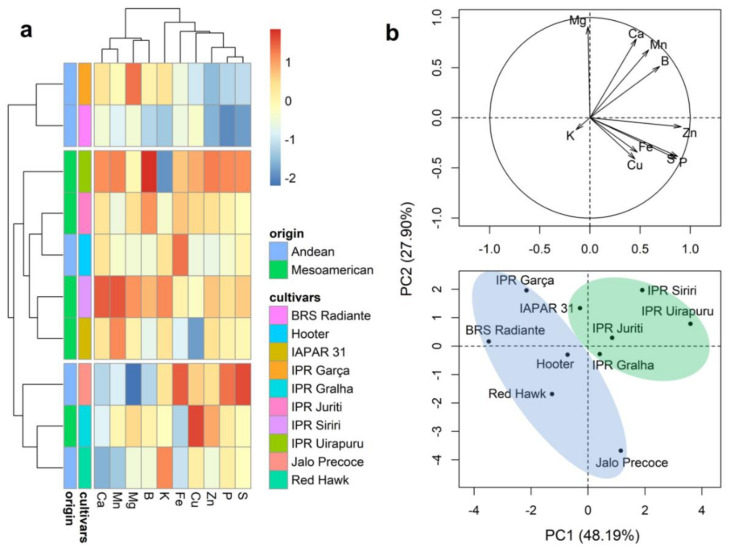
A dendrogram obtained based on Ward’s method (**a**) and principal component analysis (**b**) of 10 bean (*Phaseolus vulgaris* L.) cultivars evaluated for the contents of 10 minerals in grain cotyledons.

**Figure 5 plants-10-01246-f005:**
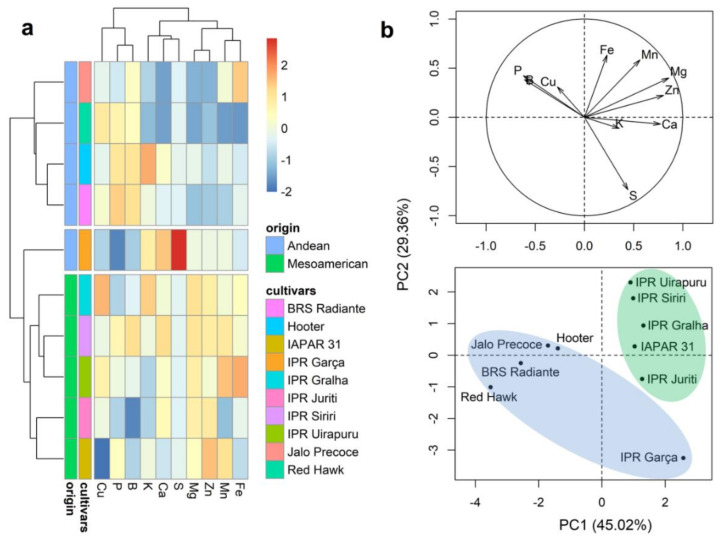
A dendrogram obtained based on Ward’s method (**a**) and principal component analysis (**b**) of 10 bean cultivars (*Phaseolus vulgaris* L.) evaluated for the contents of 10 minerals in the embryonic grain axis.

**Figure 6 plants-10-01246-f006:**
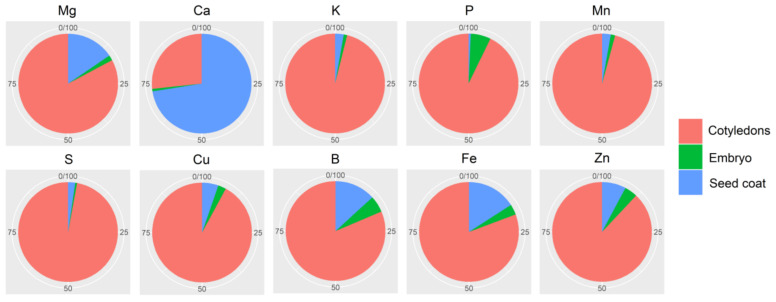
Mean percentage contents of 10 minerals in the seed coat, cotyledon, and embryonic axis of the grains of 10 bean (*Phaseolus vulgaris* L.) cultivars derived from different centers of origin.

**Table 1 plants-10-01246-t001:** Analysis of variance (ANOVA), mean values, and standard deviation of the characteristics 100-grain mass (M100), protein content (Prot), and contents of 10 evaluated minerals in the whole grains and constituent tissues (seed coat, cotyledons, and embryonic axis) of 10 bean (*Phaseolus vulgaris* L.) cultivars derived from different centers of origin.

Characteristics	Average ± Standard Deviation				F Test-ANOVA		
	Whole Grain	Seed Coat	Cotyledons	Embryonic Axis	Cultivar (C)	Tissue (T)	C × T
Mass of 100 grains (g)	38.220 ± 10.920	–	–	–	**	–	–
Protein content (%)	17.810 ± 1.480	–	–	–	**	–	–
Mg (g kg^−1^)	1.827 ± 0.119	2.396 ± 0.116	1.783 ± 0.139	1.208 ± 0.112	**	**	*
Ca (g kg^−1^)	0.656 ± 0.098	5.625 ± 1.051	0.191 ± 0.021	0.406 ± 0.035	**	**	**
K (g kg^−1^)	17.731 ± 0.723	5.100 ± 1.300	15.759 ± 3.134	12.576 ± 0.508	**	**	**
P (g kg^−1^)	2.817 ± 0.188	0.226 ± 0.043	2.871 ± 0.137	4.262 ± 0.556	**	**	*ns*
S (g kg^−1^)	2.679 ± 0.236	0.749 ± 0.121	2.885 ± 0.143	1.288 ± 0.262	*ns*	**	*ns*
Mn (mg kg^−1^)	13.712 ± 2.341	4.303 ± 0.511	14.563 ± 2.052	16.020 ± 0.611	**	**	**
Cu (mg kg^−1^)	9.269 ± 0.981	5.690 ± 0.612	9.443 ± 0.823	19.151 ± 1.363	**	**	*
B (mg kg^−1^)	7.238 ± 0.422	11.110 ± 0.646	6.473 ± 0.313	31.213 ± 3.592	**	**	*
Fe (mg kg^−1^)	42.331 ± 2.871	81.134 ± 24.196	37.492 ± 2.630	112.811 ± 8.292	**	**	**
Zn (mg kg^−1^)	22.960 ± 4.277	20.375 ± 8.613	22.351 ± 2.363	73.945 ± 6.627	**	**	**

ns, * and **: not significant, significant at the 5% and 1% probability level by F test, respectively.

**Table 2 plants-10-01246-t002:** Names, color of seed coat, origin, and breeding institutions of the 10 bean cultivars (*Phaseolus vulgaris* L.) were evaluated in this study.

Cultivar	Color of Seed Coat	Origin	Institution ^1^
IPR Siriri	Cream with brown stripes	Mesoamerican	IDR-Paraná
IPR Juriti	Cream with brown stripes	Mesoamerican	IDR-Paraná
IPR Uirapuru	Black	Mesoamerican	IDR-Paraná
IPR Gralha	Black	Mesoamerican	IDR-Paraná
IAPAR 31	Cream with brown spots	Mesoamerican	IDR-Paraná
IPR Garça	White	Andean	IDR-Paraná
Hooter	Cream with red stripes	Andean	Monsoy Ltd.a.
BRS Radiant	Cream with red stripes	Andean	Embrapa
Jalo Precoce	Yellow	Andean	Embrapa
Red Hawk	Red	Andean	MSU

^1^ IDR-Paraná (Paraná Rural Development Institute), Embrapa (Brazilian Agricultural Research Corporation), and MSU (Michigan State University).

## Data Availability

The data presented in this study are available on request from the corresponding author.

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
