# Peer review of "Genetic Variability of Mineral Content in Different Grain Structures of Bean Cultivars from Mesoamerican and Andean Gene Pools"

_plants, 2021, doi:10.3390/plants10061246_

Round 1

Reviewer 1 Report

This paper reports research on minerals and proteins contents in 10 bean cultivars and their distribution in the grain seed coat, embryo and embryonic axis. Bean genotypes of Mesoamerican and Andean origin were studied to evaluate genetic variability. The work was well executed and the interpretation of results is good. However, there are some limitations that need consideration.

Abstract:

Provide one or two sentences justifying the importance of the study.

Introduction:

Line 40:

Please provide some sentences to explain how the different bean tissues “differ with respect to the bioavailability of mineral nutrients”. This information is too general for the reader. As stated in the line 48, tissues differ with the respect to minerals contents. Why is the distribution of minerals important for human nutrition if the whole beans are usually consumed or for some purposes beans without the seed coat are used? You mentioned bioavailability of Fe from different tissues in the discussion section. How is with the bioavailability of other minerals from different tissues of cooked bean grains?

Lines 53-59: The content of this paragraph should be improved to emphasize on the problem statements prior to the objectives of the current study. In addition, the novelty of the study must be highlighted. It seems that the paper published by Ribeiro et al. (2012) (reference 13) investigated similar topic. Please highlight the novelty of the study.

Results

Table 1:

Use symbols for the minerals.

Proteins are the major bean constituents. Their contents in seed coat, cotyledons and embryonic axis were not determined. Any rationale for this?

Line 69: Check the value of P content in embryonic axis in the Table 1 and in the text.

Line 70: Probably Fe instead of S?

Discussion:

Figure 5, line 179: delete “cultivars”

Lines 229-230: Any explanation for lower concentrations of minerals in the grains of bigger size.

Line 266: Is S crucial for the synthesis of all essential amino acids?

Line 278: delete “are”

Materials and methods

Line 321: HCl instead HCL

Lines 328-329: use just symbols for elements

Conclusion:

Do not reiterate the results obtained. Conclude what new information was obtained from the study and how the finding advances our knowledge in the field.

Table A2: Use symbol for elements and report the content as (g kg-1 DM). Check the letters; in some cases (Ca, P) there are two lowercase letters. The same comment is also for Cu in Table A3.

Table A3: Use symbol for elements and report the content as (mg kg-1 DM).

References:

Some of the references cited are fairly old (37-39). More recent publications should be referred.

Author Response

[1] Abstract: Provide one or two sentences justifying the importance of the study.

Answer: A sentence about the importance of this type of study is inserted.

[2] Introduction: Line 40: Please provide some sentences to explain how the different bean tissues “differ with respect to the bioavailability of mineral nutrients”. This information is too general for the reader. As stated in the line 48, tissues differ with the respect to minerals contents. Why is the distribution of minerals important for human nutrition if the whole beans are usually consumed or for some purposes beans without the seed coat are used? You mentioned bioavailability of Fe from different tissues in the discussion section. How is with the bioavailability of other minerals from different tissues of cooked bean grains?

Answer: We agree with that suggestion and this sentence has been improved.

[3] Lines 53-59: The content of this paragraph should be improved to emphasize on the problem statements prior to the objectives of the current study. In addition, the novelty of the study must be highlighted. It seems that the paper published by Ribeiro et al. (2012) (reference 13) investigated similar topic. Please highlight the novelty of the study.

Answer: We agreed and a paragraph has been inserted.

[4] Table 1: Use symbols for the minerals.

Answer: We agree with that suggestion and the chemical symbols has been inserted.

[5] Proteins are the major bean constituents. Their contents in seed coat, cotyledons and embryonic axis were not determined. Any rationale for this?

Answer: We agree and this point will be considered in future studies.

[6] Line 69: Check the value of P content in embryonic axis in the Table 1 and in the text.

Answer: This sentence has been corrected.

[7] Line 70: Probably Fe instead of S?

Answer: This sentence has been corrected.

[8] Discussion: Figure 5, line 179: delete “cultivars”

Answer: This word has been deleted.

[9] Lines 229-230: Any explanation for lower concentrations of minerals in the grains of bigger size.

Answer: No. Unfortunately, there is no consensus among researchers about this.

[10] Line 266: Is S crucial for the synthesis of all essential amino acids?

Answer: This sentence has been improved

[11] Line 278: delete “are”

Answer: This word has been deleted.

[12] Materials and methods; Line 321: HCl instead HCL

Answer: This term has been corrected.

[13] Lines 328-329: use just symbols for elements

Answer: We agree with that suggestion and the chemical symbols has been inserted.

[14] Conclusion: Do not reiterate the results obtained. Conclude what new information was obtained from the study and how the finding advances our knowledge in the field.

Answer: We agree with that suggestion and the Conclusion section has been improved.

[15] Table A2: Use symbol for elements and report the content as (g kg-1 DM). Check the letters; in some cases (Ca, P) there are two lowercase letters. The same comment is also for Cu in Table A3.

Answer: We agree with that suggestion and the chemical symbols has been inserted.

[16] Table A3: Use symbol for elements and report the content as (mg kg-1 DM).

Answer: We agree with that suggestion and the chemical symbols has been inserted.

[17] References: Some of the references cited are fairly old (37-39). More recent publications should be referred.

Answer: Although old, these references are related to statistical methods/tests. Thus, there is no possibility of replacing them with recent references.

Reviewer 2 Report

This is an outstanding piece of research. I have nothing to say regarding the science. However, I wanted to recommend that you consider the 10% of male scientists who are color blind. (I am not color blind, but I personally know eminent scientists who are.) Figure 1 is unreadable for those individuals who are red/green color blind, which is the most common type of color blindness. Similarly, embryos and cotyledons in figure 6 are also indistinguishable for this disability. A blue/red or blue/orange color scheme is better for most color blind persons. Personally, I feel that if you cannot print a paper on a gray scale and be able to read it, then the figures need adjustment in this regard. I don't expect that you will change all the figures for this paper, but please consider this aspect of publishing with future papers. I don't mean to harp on this too much. This is an excellent paper. The science and the writing are of the highest standards. I enjoyed reading the paper and learned much from it.

Author Response

We agree with this question related to colors. Unfortunately, changes in the graphs in this article will require major changes in the presentation of results. However, we will consider this problem related to colors in future studies. Thank you in advance for understanding.